# The Potential Geographic Distribution of *Bactrocera minax* and *Bactrocera tsuneonis* (Diptera: Tephritidae) in China

**DOI:** 10.3390/insects16121277

**Published:** 2025-12-16

**Authors:** Yunfa Wan, Chuanren Li, Zhengping Yin, Zailing Wang

**Affiliations:** 1Department of Entomology, Yangtze University, Jingzhou 404023, China; 18771411205@163.com (Y.W.); 13986706558@163.com (C.L.); 2Science and Technology Bureau, Yidu Municipal People’s Government, Yichang 443000, China; 13986768503@163.com

**Keywords:** citrus pest, *Bactrocera minax*, *Bactrocera tsuneonis*, MaxEnt, climate change, environmental variables

## Abstract

This study analyzed the relationship between the suitable habitats of *Bactrocera minax* (Enderlein) (Diptera: Tephritidae) and *Bactrocera tsuneonis* (Miyake) (Diptera: Tephritidae) in China, and various environmental factors using the MaxEnt ecological niche model. By incorporating citrus plant coverage as a key environmental variable, the model achieved higher accuracy and improved predictive performance compared to previous studies. At present, *B. minax* and *B. tsuneonis* are primarily distributed in southern Chinese provinces, where citrus cultivation is widespread. The spatial distribution of citrus plants plays a pivotal role in shaping the geographical range of these fruit flies. Across all climate scenarios, *B. tsuneonis* exhibits a consistent trend of habitat contraction. Higher greenhouse gas emission scenarios appear to contribute to a reduction in *B. minax* and *B. tsuneonis*, potentially aiding in pest control. Nevertheless, regions engaged in citrus production—particularly eastern Sichuan—should implement enhanced management measures to prevent the invasion and spread of *B. minax* and *B. tsuneonis*. These findings provide valuable scientific insights for forecasting species distributions and developing targeted management and control strategies. The results support the sustainable and resilient development of China’s citrus industry.

## 1. Introduction

Tephritidae is one of the largest families of Diptera, comprising more than 4000 fruit fly species, among which approximately 350 are considered economically important pests [1]. The *Tetradacus* is the subgenus of *Bactrocera* (family Tephritidae), widely distributed across tropical and subtropical regions of East Asia [2,3]. *Bactrocera minax* and *Bactrocera tsuneonis* are the only two species classified under *Tetradacus*.

*B. minax*, commonly known as the ‘citrus maggot fly’ in China, is a devastating pest of citrus fruits [4]. The larvae feed on orange segments and seeds, leading to internal decay of citrus fruits [5]. *B. tsuneonis* is a univoltine tephritid pest [6,7], that infests citrus species including sweet orange, grapefruit, lemon, and other citrus fruit trees [8]. As an important quarantine pest both internationally and domestically, it is officially listed in the List of Imported Phytosanitary Pests of the People’s Republic of China and China’s National List of Agricultural Plant Quarantine Pests (2024) [9,10].

The two pests share almost identical morphological characteristics and occupy highly overlapping ecological niches. *B. tsuneonis* can be distinguished from *B. minax* by its 1–2 pairs of anterior supra-alar seta on the mesonotum (absent in *B. minax*) and a female ovipositor about half the abdominal length (as long as the abdomen in *B. minax*) [4,11]. In addition, the two pests can be accurately identified through DNA barcoding [12,13]. They have similar damage symptoms. Infested immature fruits undergo premature yellowing, followed by rotting or loss of edible value [14], severely threatening the profitability of citrus cultivation in China. They are primarily distributed in Yunnan, Sichuan, Guizhou, Hunan, and Guangdong. Both pests primarily rely on short-distance flight or medium-range wind-assisted dispersal. Research indicates that *B. minax* can spread up to 1500 m when aided by favorable wind [15]. However, long-distance dispersal is primarily achieved through the transportation of seedlings and fruits.

*Citrus reticulata* as the primary host of *B. minax* and *B. tsuneonis*, represents one of the world’s most widely cultivated and economically important crop groups [16]. Since the beginning of this century, both the planting area and production of citrus in China have increased steadily. However, significant challenges remain in citrus pest prevention and control [17], particularly following the 2008 fruit maggot outbreak in Guangyuan, Sichuan, which caused more than losses exceeding 20 billion yuan and severely impacted the industry’s reputation [18,19]. Therefore, accurate species identification and precise monitoring are crucial for effective pest management. Modeling the potential distribution of suitable habitats of *B. minax* and *B. tsuneonis* forms the foundation for safeguarding the citrus industry in China.

The Maximum Entropy model (MaxEnt), which integrates machine learning with the principle of maximum entropy, is widely recognized as one of the most robust tools for predicting species’ potential distributions. It is characterized by high prediction accuracy and standardized input formats, and is extensively used in fields such as biogeography, invasion biology, conservation biology, and studies on the impacts of global climate change on species distributions [20]. For example, Fu et al. predicted that *B. minax* potential suitable habitats may shift toward the lower–middle Yangtze River Basin under future climates [21]. Similarly, Mao employed the MaxEnt model to predict that, under future climate change scenarios, *B. tsuneonis* is likely to expand its potential suitable range into northern and western China [22].

However, citrus distribution has not previously been incorporated into distribution models of *B. minax* and *B. tsuneonis*, and a systematic comparison of the suitable habitats of *B. minax* and *B. tsuneonis* remains lacking. Building on previous research, this study employs the MaxEnt model, integrating citrus plant coverage as a key environmental variable alongside geographic occurrence data and climate data from different periods to simulate the PGDs of *B. minax* and *B. tsuneonis* species in China. It examines current patterns of suitable habitat distribution and projects future shifts in PGDs under various climate change scenarios, offering both empirical data and a theoretical foundation for safeguarding the citrus industry.

## 2. Materials and Methods

### 2.1. Occurrence Record of B. minax and B. tsuneonis

The distribution data for *B. minax* and *B. tsuneonis* were obtained from online databases and field surveys. Relevant occurrence records of *B. minax* and *B. tsuneonis* were downloaded from the Global Biodiversity Information Facility (GBIF Occurrence Download: https://doi.org/10.15468/dl.5ugbmt, accessed on 7 November 2024) [23], the Centre for Agriculture and Biosciences International (CABI, http://www.cabi.org, accessed on 23 April 2025), and the National Animal Specimen Resource Center (http://museum.ioz.ac.cn/specimens.aspx?str=Arthropoda&type=1, accessed on 23 April 2025). Field sampling was conducted in Yunnan, Guizhou, Sichuan, Chongqing, Hunan, Guangxi, and Hubei provinces. After removing duplicate entries and records lacking precise geographic information, a total of 91 valid occurrence points for *B. minax* and 43 for *B. tsuneonis* were retained.

### 2.2. Environment Variables Related to B. minax and B. tsuneonis

A total of 24 environmental variables were incorporated into the modeling process, including elevation, slope, aspect, 19 bioclimatic variables, the Normalized Difference Vegetation Index (NDVI), and a citrus distribution index. Ultimately, five bioclimatic variables (BIO2, BIO3, BIO4, BIO6, and BIO14), three topographic variables (elevation, slope, and aspect), and one host-related variable (citrus cover index) were selected for model construction. To ensure comparability of model outputs across different climate scenarios, elevation, slope, aspect, and citrus cover were assumed to remain constant over time.

### 2.3. Climatic Data

Contemporary (1970–2000) climate data were obtained from the WorldClim database (https://www.worldclim.org/, accessed on 23 April 2025), including 19 bioclimatic variables (BIO1–BIO19) related to temperature and precipitation, with a spatial resolution of 2.5 arc-minutes [24].

Future climatic projections were obtained from the BCC-CSM2-MR model, developed by the Beijing Climate Center, which is part of the Coupled Model Intercomparison Project Phase 6 (CMIP6). Four Shared Socioeconomic Pathway (SSP) scenarios were used to model potential future distributions for the period 2041–2080: SSP1–2.6 (low forcing), SSP2–4.5 (moderate forcing), SSP3–7.0 (moderate to high forcing), SSP5–8.5 (high forcing).

### 2.4. Topographic Data

Vegetation data, Normalized Difference Vegetation Index (NDVI) values, were obtained from the Distributed Active Archive Center for Biogeochemical Dynamics (DAAC, https://daac.ornl.gov/, accessed on 23 April 2025, Oka Ridge, TN, USA). The multi-year average NDVI was calculated using the maximum value composite method.

Given that *B. minax* and *B. tsuneonis* are oligophagous pests mainly infesting citrus, occurrence data for citrus plants were also downloaded from GBIF (GBIF Occurrence Download: https://doi.org/10.15468/dl.8w2g5u, accessed on 29 March 2025) [25]. Following the methods of Zhao and Crego, 10,000 background (pseudo-absence) points were randomly generated across China [26,27,28]. Kriging interpolation was applied to create a citrus distribution raster [29], which was included as a key environmental variable.

### 2.5. Distribution Model Construction and Accuracy Evaluation

All environmental variables were processed and defined using ArcGIS 10.8. The geographic coordinate system was set to CGCS_WGS_1984, and the projected coordinate system was set to GCS Krasovsky 1940–Albers. Environmental data within China were extracted in bulk, resampled to a spatial resolution of 8 km × 8 km, and subsequently converted to ASCII format for use in MaxEnt modeling.

During the modeling process, 75% of the *B. minax* and *B. tsuneonis* occurrence data were randomly selected as training dataset, while the remaining 25% were used as test dataset. The maximum number of iterations was set to 10,000, the number of replicates was set to 10, and all other parameters were kept at their default settings [30].

To minimize multicollinearity among environmental variables and prevent overfitting in the MaxEnt model, ENMTools 3.4.4 [31] was employed to preprocess environmental variables and species occurrence data. Correlation analysis was conducted in R to address potential multicollinearity among the 19 bioclimatic variables, as high multicollinearity can lead to overfitting and reduced predictive accuracy [32,33]. When the Pearson correlation coefficient between two variables exceeded 0.8, the variable with the higher contribution rate to the distribution of *B. minax* and *B. tsuneonis* were preferentially retained [34,35].

### 2.6. Division of the PGDs and Calculation of Centroids

The classification of climate suitability refers to the classification criteria outlined for assessment likelihood in the 6th Assessment Report of the Intergovernmental Panel on Climate Change (IPCC) [36]. Regions were then classified based on species richness from highest to lowest, using the spatial analyst tools in ArcGIS 10.8 and applying the natural breaks (Jenks) method [37]. The PGDs of *B. minax* and *B. tsuneonis* under current and future climate conditions were categorized based on probability values: unsuitable habitat (0–0.1), low-suitability habitat (0.1–0.3), moderate-suitability habitat (0.3–0.5) and high-suitability habitat (0.5–1). The areas of each PGD at different periods were calculated by using the spatial statistical function of ArcGIS. Additionally, the centroid coordinates of the PGDs were computed using R 4.3.1 software. Changes in centroid positions across scenarios were analyzed to predict the spatiotemporal distribution trends of *B. minax* and *B. tsuneonis* [38].

## 3. Results

### 3.1. Current Distribution of B. minax and B. tsuneonis

The occurrence of *B. minax* is densely concentrated in Guizhou and Hunan provinces, with sparser distributions observed in Hubei, Sichuan, Yunnan, Guangxi, Guangdong, Chongqing, and Shanxi (Figure 1A). In contrast, *B. tsuneonis* exhibits a denser but more geographically restricted distribution compared to *B. minax* (Figure 1B).

### 3.2. Model Prediction Result Verification

The AUC value of MaxEnt model is 0.969, which reaches an excellent level, indicating that the MaxEnt model can well predict the distribution of suitable habitat of *B. minax* and *B. tsuneonis* in China.

### 3.3. Evaluation of Important Environment Variables

The Citrus distribution index, BIO 14, BIO 6, and elevation contributed 39.9%, 14.3%, 9.2%, and 8%, respectively, with a cumulative contribution of 71.4% (Table 1). *B. minax* was found to favor areas where the citrus distribution index is ≥0.28, the precipitation during the driest month ranges from 18 to 63 mm, the mean temperature of the coldest month is between 1.4 °C and 5 °C, and elevation ranges from approximately 150 to 625 m. In contrast, *B. tsuneonis* favors habitats where the citrus distribution index is ≥0.45, the precipitation during the driest month ranges from 19 to 60 mm, the mean temperature of the coldest month is between 2 °C and 7 °C, and elevation ranges from approximately 240 to 970 m (Figure 1 and Figure 2).

### 3.4. PGDs of B. minax and B. tsuneonis in China

The PGDs of both pests are predominantly confined to southern China, primarily south of the 0 °C isotherm for the coldest month, with small patches of low-suitability habitat identified in southern Tibet. No suitable habitats were detected in the Poyang Lake Basin or Hainan. Currently, the suitable habitat of *B. minax* is primarily concentrated in southern China, particularly in the middle and lower Yangtze River regions, the Yunnan-Guizhou Plateau, and the southeastern hills, with smaller scattered patches in Guangdong and Guangxi (Figure 3A). The suitable habitat of *B. tsuneonis* is broader in extent, covering large areas along the lower Yangtze River, Guangdong, and Guangxi (Figure 3B). High-suitability habitats for both species are primarily distributed from the eastern Yunnan-Guizhou Plateau and the upper Yuan River basin, extending in two distinct directions: for *B. minax*, eastward along the plateau and the upper Yuan River, following the 20 mm precipitation contour for the driest month along the lower Yangtze; for *B. tsuneonis*, westward into Guizhou, Chongqing, and eastern Sichuan, with additional scattered pockets along the eastern coastal regions.

Nationally, the total suitable habitat for *Bactrocera minax* is estimated at approximately 1,360,000 km^2^, representing about 14% of China’s land area. This includes 989,000 km^2^ of low-suitability habitat, 394,000 km^2^ of moderate-suitability habitat, and 148,000 km^2^ of high-suitability habitat. *B. tsuneonis* exhibits a broader distribution, with suitable habitat covering approximately 1,870,000 km^2^ (around 19% of China), comprising 1,095,000 km^2^ of low-suitability habitat, 452,000 km^2^ of moderate suitability, and 330,000 km^2^ of high suitability. These findings indicate that *B. minax* and *B. tsuneonis* have a broad ecological amplitude, with *B. tsuneonis* occupying a significantly larger potential range than *B. minax*. Comparison of high- and low-suitability areas reveals that zones of high-suitability overlap at the Hunan–Guizhou border, after which the distributions diverge—extending eastward for *B. minax* and westward for *B. tsuneonis* (Figure 3C). Their low-suitability habitats form a distinct ‘C’-shaped belt encompassing eastern, southern, and western China, eventually merging into moderate and high-suitability zones in eastern Sichuan and southern Chongqing (Figure 3D). Additionally, isolated pockets of suitable habitat for both species were identified in Medog County, Nyingchi, Tibet.

### 3.5. PGDs of B. minax and B. tsuneonis Under Future Climate Scenarios

Under four future climate scenarios (SSP1–2.6, SSP2–4.5, SSP3–7.0, and SSP5–8.5), both *B. minax* and *B. tsuneonis* exhibit overall declines in climatic suitability (Figure 4). Under the SSP1–2.6 scenario, the total suitable habitat for *B. minax* expands slightly, with high-suitability areas increasing by approximately 27% during 2061–2080, particularly in eastern Guizhou, western Hunan, southern Hubei, western Anhui, and central Sichuan. In contrast, *B. tsuneonis* shows a contraction in suitable habitat, with moderate-suitability areas decreasing by 34% by 2061–2080, mainly in southern Chongqing, Guizhou, and western Hunan. Under the SSP2–4.5 scenario, *B. minax* initially exhibits a modest increase in total suitable habitat from 2041 to 2060, followed by a subsequent decline. Meanwhile, *B. tsuneonis* experiences continuous habitat contraction across low-, moderate-, and high-suitability classes, progressively retreating toward a core distribution zone centered in Guizhou. In the SSP3–7.0 scenario, both species undergo reductions in total suitable habitat during 2041–2060, followed by a partial rebound in 2061–2080, primarily attributable to shifts in moderate-suitability zones. However, *B. tsuneonis* experiences a substantial net reduction during 2041–2060, driven by losses in both low- and high-suitability habitats. Similarly, under the high-emission SSP5–8.5 scenario, *B. tsuneonis* continues to lose suitable habitat as both low- and high-suitability areas shrink. In contrast, *B. minax* shows a slight increase in habitat suitability during 2041–2060, mainly due to the expansion of low-suitability areas, followed by a notable decline by 2061–2080 (Figure 5).

### 3.6. Centroid Shift in B. minax and B. tsuneonis Under Future Climate Change

The projected shifts in the geographic centroids of suitable habitats for *B. minax* and *B. tsuneonis* under various climate change pathways are depicted in Figure 6. Under current climatic conditions, the centroid for *B. minax* is currently situated in southwestern Zhangjiajie City, Hunan Province. Under the SSP1–2.6 pathway, it initially shifts northeastward and then southeastward, ultimately stabilizing in Changde City, Hunan Province. Under the SSP2–4.5 pathway, the centroid first moves northeastward before shifting southwestward. In the SSP3–7.0 pathway, it initially shifts southward before trending eastward. Under the SSP5–8.5 high-emission pathway, the centroid gradually shifts eastward and ultimately stabilizes in Changde City (Figure 6A). For *B. tsuneonis*, the current centroid is located in Huaihua City, Hunan Province. Under the SSP1–2.6 pathway, the centroid initially shifts northeastward and then westward, ultimately stabilizing in the Xiangxi Tujia and Miao Autonomous Prefecture. Under both the SSP2–4.5 and SSP3–7.0 pathways, the centroid first shifts westward and subsequently trends northward. Under SSP5–8.5, the centroid exhibits a northwestward shift followed by a northeastward movement, eventually stabilizing in the Xiangxi Tujia and Miao Autonomous Prefecture (Figure 6B).

## 4. Discussions

*B. minax* and *B. tsuneonis* are oligophagous pests that exclusively infest citrus species [39]. This study evaluates the effects of abiotic factors—including climate, topography, and vegetation cover—on their distribution. It further compares the suitable habitat ranges of both species, analyzes changes under various climate scenarios, and tracks spatial shifts in their distribution centroids. Results indicate that both species predominantly occupy regions south of the 0 °C isotherm during the coldest month. This contrasts with previous studies, which identified suitable habitats in non-citrus-growing areas such as Poyang Lake [22,26], and conversely, classified actual infestation areas such as Guangxi and Guangdong as unsuitable areas [21]. These discrepancies highlight substantial deviations from observed distributions. Moreover, this study identifies low-suitability habitats for both species in Medog County, Nyingchi City, Tibet. Situated on the eastern slopes of the Himalayas, Medog experiences a subtropical humid climate with mild winters, abundant summer rainfall, and an annual average temperature of 18–22 °C—conditions favorable for citrus cultivation [40]. By incorporating a rasterized map of citrus distribution in China (Appendix A) as a key environmental variable in the model, the study enhances the predictive accuracy of the distribution of these pests. It is crucial to implement targeted monitoring and control strategies for these fruit fly species in agricultural practice with particular attention to *B. tsuneonis*, to mitigate their impact on citrus production.

The distribution index of citrus plants is the primary environmental determinant influencing the suitable habitat distribution of *B. minax* and *B. tsuneonis*. Insects rely on specific hosts for survival and reproduction, and their distribution ranges are largely determined by host plant availability. Similar patterns have been observed, including the strong correlation between the distribution of Agromyzidae and host plant presence in Ningxia [41]. Both *B. minax* and *B. tsuneonis* are oligophagous pests dependent on citrus as their host, resulting in a distribution pattern closely aligned with citrus cultivation areas. Given that citrus is widely planted across southern China, *B. minax* and *B. tsuneonis* are consequently more prevalent in these regions. However, in certain citrus-growing areas such as Hainan, Beijing, and southern Yunnan, *B. minax* and *B. tsuneonis* were not predicted to find suitable habitats. These regions, therefore, require enhanced quarantine measures to prevent potential invasions.

Precipitation and temperature are key environmental factors influencing the occurrence of *B. minax* and *B. tsuneonis*. Precipitation affects pupal ground development (PGDs) indirectly by altering soil moisture. Zhang found that soil moisture at a depth of 12 cm favored the eclosion of *B. minax*, whereas conditions at 25 cm were unfavorable [42]. Subsequent studies by Lv suggested that moderate winter rainfall, corresponding to soil water content of 10–15%, supports the overwintering and eclosion of *B. minax* pupae [43], whereas excessively dry or wet conditions adversely affect eclosion rates and timing [44]. The minimum temperature of the coldest month (BIO 6) emerged as the third most influential factor for the distribution of *B. minax* and *B. tsuneonis*. Overwintering temperatures play a critical role, as they affect insect emergence in the following year [33]. Zhuo demonstrated that overwintering eggs of *Apolygus lucorum* (Hemiptera: Miridae) exit diapause completely after 65 days of exposure to 2 °C, with hatching rates positively correlated with the duration of cold treatment [45]. Similarly, Zhou found that low temperatures delay the development of *B. minax* diapause pupae, thereby ensuring synchronized emergence with food availability and improving survival rates [46]. In China, elevated temperatures may shorten the pupal diapause, potentially causing early emergence, food scarcity, and increased mortality. Luo and Chen identified temperature as the key ecological factor regulating *B. minax* pupal development [11]. A subsequent study estimated the developmental threshold temperature at 10.57 °C [47].

Topographic factors, particularly elevation, also indirectly influence PGDs. In China, *B. minax* and *B. tsuneonis* predominantly inhabit hilly regions at elevations ranging from 150 to 900 m [48]. High-altitude regions restrict their development due to low temperatures, while lowland plains often host smaller populations, likely due to intensive agricultural practices and the presence of natural enemies. In contrast, complex mountainous terrain provides sheltered microhabitats, including forests and weedy patches, which support adult survival.

Under future climate scenarios, projections from multiple models indicate an overall reduction in suitable habitat for *B. minax* and *B. tsuneonis*. Notably, under the SSP2–4.5 scenario, the suitable habitat for *B. minax* is projected to expand during the 2040–2060 period. This projected expansion underscores the importance of proactive monitoring and preventive measures in regions where habitat suitability is anticipated to rise. In contrast, projections for *B. tsuneonis* across all four Shared Socioeconomic Pathways (SSPs) suggest a gradual decline across high, moderate, and marginal suitability classes. Given its designation as a major quarantine pest in both domestic and international contexts, increased surveillance is essential throughout production and distribution processes. In eastern Sichuan, a particularly sharp transition is observed, as highly suitable habitat for *B. tsuneonis* shifts westward into a narrow zone of low suitability in the absence of an intermediate zone of moderate suitability. This abrupt transition could elevate the risk of localized invasion, consistent with the findings of Zhao [49]. Interestingly, under the high-emission SSP5–8.5 scenario, the contraction of moderate and high-suitability habitat is less pronounced than in other scenarios, suggesting that the impact of emission levels on the distribution of *B. minax* and *B. tsuneonis* may be relatively limited.

Changes in the citrus distribution index and the bioclimatic variable BIO14 under climate change play a critical role in shaping the potential geographic distributions (PGDs) of *B. minax* and *B. tsuneonis*. However, the citrus distribution index is projected to remain relatively stable over time. As a result, BIO14 emerges as the dominant factor driving shifts in the distributional centroids of these species. To elucidate the relationship between habitat suitability changes and BIO14, we mapped the 20 mm precipitation isoline for BIO14 under both current and future climate scenarios (Appendix A). Across all four climate scenarios, this isoline exhibits only minor spatial variation. Consequently, the geographic centroids of *B. minax* and *B. tsuneonis* exhibit minimal displacement. This relative stability is largely attributable to the stable distribution of citrus cultivation zones, which are closely associated with the presence of *B. minax* and *B. tsuneonis* and play a defining role in their geographic range. Therefore, enhanced orchard management strategies in China’s citrus-producing regions are essential to prevent the establishment and spread of *B. minax* and *B. tsuneonis* populations.

This study analyzed the relationship between the suitable habitats of *B. minax* and *B. tsuneonis* in China and various environmental factors using the MaxEnt ecological niche model. By incorporating citrus plant coverage as a key environmental variable, the model achieved higher accuracy and improved predictive performance compared to previous studies. These findings provide valuable scientific insights for forecasting species distributions and developing targeted management and control strategies. At present, *B. minax* and *B. tsuneonis* are primarily distributed in southern Chinese provinces, where citrus cultivation is widespread. The spatial distribution of citrus plants plays a pivotal role in shaping the geographical range of these fruit flies. Across all climate scenarios, *B. tsuneonis* exhibits a consistent trend of habitat contraction. Higher greenhouse gas emission scenarios appear to contribute to a reduction in *B. minax* and *B. tsuneonis* suitability, potentially aiding in pest control. Nevertheless, regions engaged in citrus production—particularly eastern Sichuan—should implement enhanced management measures to prevent the invasion and spread of *B. minax* and *B. tsuneonis*. However, this study does not incorporate the potential effects of adaptive evolutionary processes, nor does it account for human-mediated dispersal mechanisms, such as long-distance jump dispersal via seedling or plant material transport. The omission of these processes constitutes a significant source of uncertainty in the predictive outcomes and should be considered a priority for refinement in future research, thereby supporting the citrus industry’s sustainable and resilient development in China.

## Figures and Tables

**Figure 1 insects-16-01277-f001:**
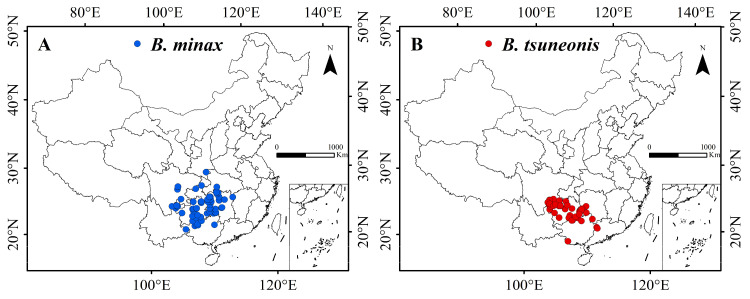
Occurrence of *B. minax* (**A**) and *B. tsuneonis* (**B**).

**Figure 2 insects-16-01277-f002:**
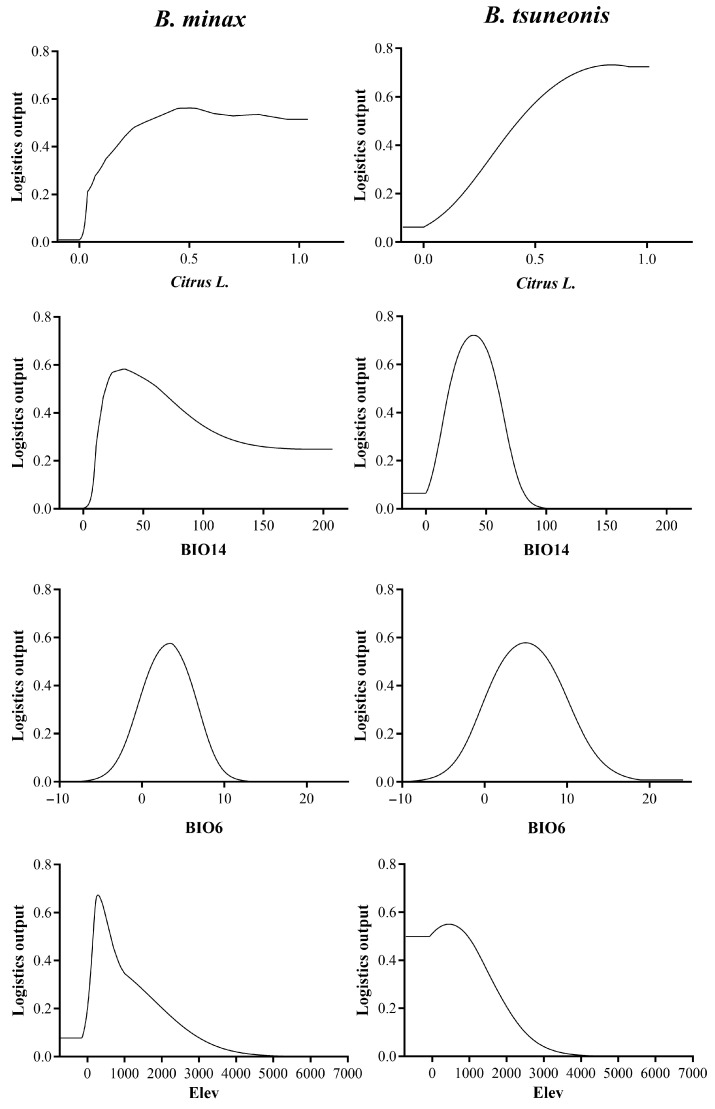
Response curve of major environmental variables of *B. minax* and *B. tsuneonis*. *Citrus* L.: response curve of citrus distribution index; BIO14: response curve of precipitation of driest month; BIO6: response curve of min temperature of coldest month; Elev: response curve of elevation.

**Figure 3 insects-16-01277-f003:**
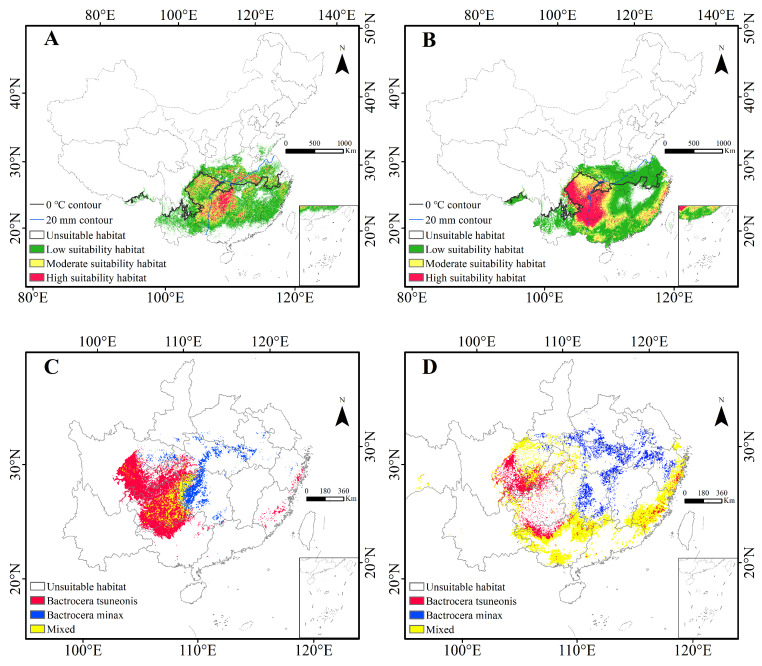
Current potential distribution of *B. minax* and *B. tsuneonis* in China. (**A**): Distribution of contemporary suitable habitat of *B. minax*; (**B**): Distribution of contemporary suitable habitat of *B. tsuneonis*; (**C**): Crossing analysis of high-suitability habits of *B. minax* and *B. tsuneonis*; (**D**): Crossing analysis of low-suitability habits of *B. minax* and *B. tsuneonis*.

**Figure 4 insects-16-01277-f004:**
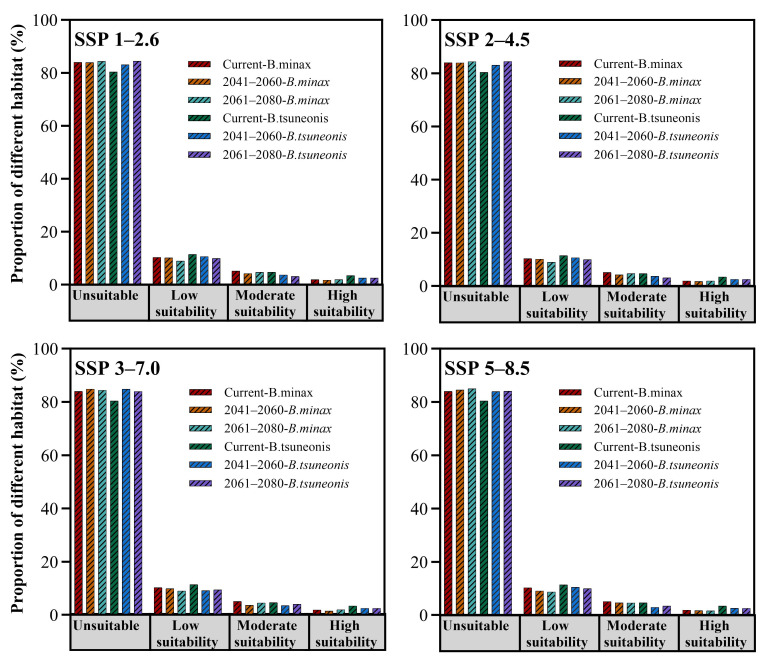
Proportion of different habitats of *B. minax* and *B. tsuneonis* for current and future climate conditions.

**Figure 5 insects-16-01277-f005:**
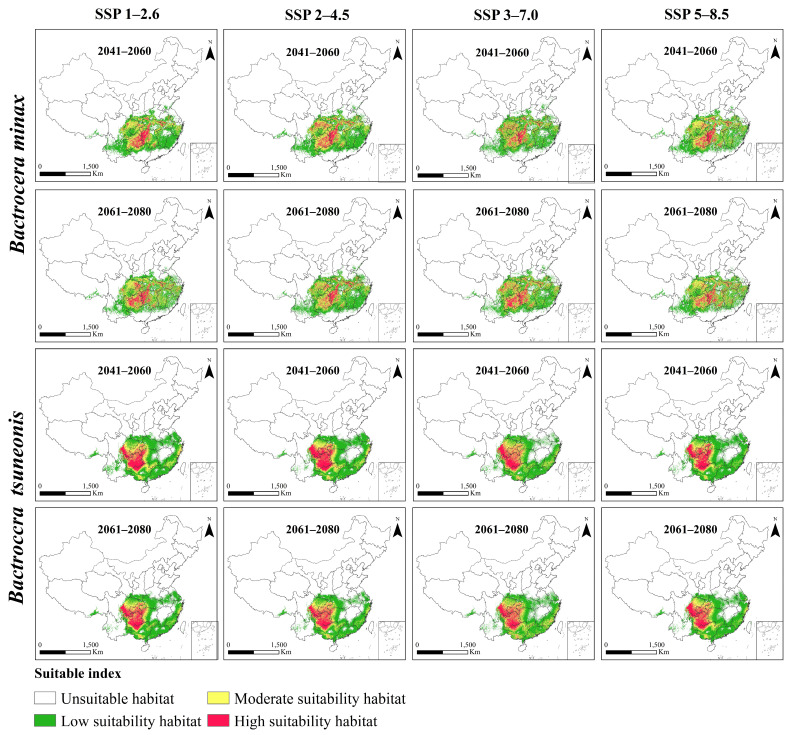
Habitats of *B. minax* and *B. tsuneonis* in China under different shared socioeconomic pathways (SSP).

**Figure 6 insects-16-01277-f006:**
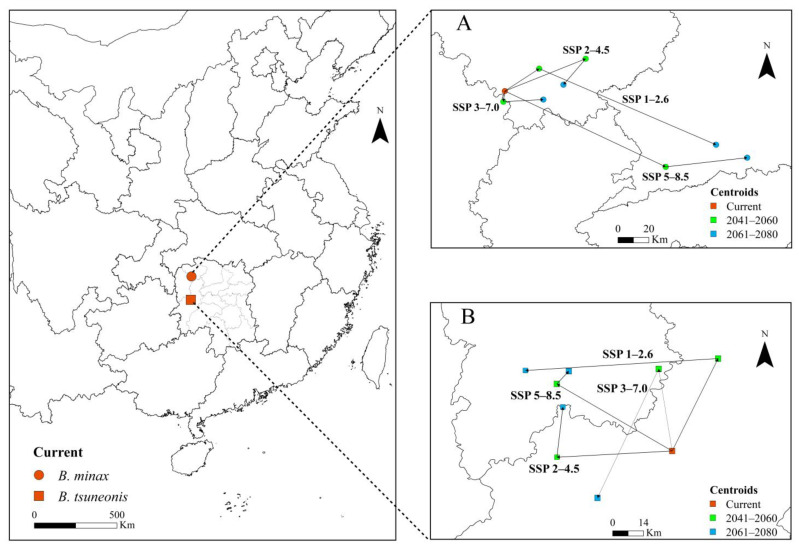
Changes in the centroid shifts in (**A**) *B. minax* and (**B**) *B. tsuneonis* in China under four shared socioeconomic pathways (SSP).

**Table 1 insects-16-01277-t001:** Selection of environmental variables and the contribution ratio.

Variable Code	Description	Contribution Rate %	Permutation Importance %
*Citrus* L.	Citrus distribution index	39.9	0.8
BIO 14	Precipitation of driest month	14.3	10.9
BIO 6	Min temperature of coldest month	9.2	14.2
Elev	Elevation (m)	8	5.4
BIO 19	Precipitation of coldest quarter (mm)	5.6	17.7
BIO 2	Mean diurnal range (°C)	2.4	2.9
BIO 3	Isothermality (BIO2/BIO7) (×100)	2.1	1.8
BIO 4	Seasonal variation in temperature	1.8	2.8
aspect	Aspect (°)	0.8	1
slope	Slope (°)	0.7	0.3

The Jackknife test was used to calculate the relative importance and contribution rate of each environmental variable.

## Data Availability

The original contributions presented in this study are included in the article/Appendix A. Further inquiries can be directed to the corresponding author.

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
