# Peer review of "The Potential Geographic Distribution of *Bactrocera minax* and *Bactrocera tsuneonis* (Diptera: Tephritidae) in China"

_insects, 2025, doi:10.3390/insects16121277_

Round 1
Reviewer 1 Report
Comments and Suggestions for Authors
The manuscript is overall well organized and clearly written, and the study is of practical value for important pests on Citrus reticulata, which is the primary host of Bactrocera minax and B. tsuneonis. The survey of two citrus maggot flies provides useful information for understanding pest status under new ecological conditions and prognosis of dispersal patterns caused by global warming scenarios. I think that authors must provide additional information about morphological discrimination even when differences are tiny. The overlapping distribution must consider an explanation of how species are treated in classical or molecular taxonomy. The author must include a few sentences in the introduction to cover problems in distinguishing these species and to cite literature references dealing with this issue (e.g., Zheng et al., 2019). New species-specific primers for molecular diagnosis of Bactrocera minax and Bactrocera tsuneonis (Diptera: Tephritidae) in China based on DNA barcodes. Insects, 10(12), 447). After this, the work is suitable for publication.
Author Response
Comments 1: The manuscript is overall well organized and clearly written, and the study is of practical value for important pests on Citrus reticulata, which is the primary host of Bactrocera minax and B. tsuneonis. The survey of two citrus maggot flies provides useful information for understanding pest status under new ecological conditions and prognosis of dispersal patterns caused by global warming scenarios. I think that authors must provide additional information about morphological discrimination even when differences are tiny. The overlapping distribution must consider an explanation of how species are treated in classical or molecular taxonomy. The author must include a few sentences in the introduction to cover problems in distinguishing these species and to cite literature references dealing with this issue (e.g., Zheng et al., 2019). New species-specific primers for molecular diagnosis of Bactrocera minax and Bactrocera tsuneonis (Diptera: Tephritidae) in China based on DNA barcodes. Insects, 10(12), 447). After this, the work is suitable for publication.
Response 1: We sincerely thank the reviewer for their positive and encouraging comments on our manuscript, and for acknowledging its practical value. We are also grateful for the constructive suggestions regarding species discrimination. In introduction section we add some knowledge and cite relative references in line 68-71. “The two pests share almost identical morphological characteristics and occupy highly overlapping ecological niches. B. tsuneonis can be distinguished from B. minax by its 1-2 pairs of anterior supra-alar seta on the mesonotum (absent in B. minax) and a female ovipositor about half the abdominal length (as long as the abdomen in B. minax)(Chen and Xie, 1955; Luo and Chen, 1987). In addition, the two pests can be accurately identified through DNA barcoding (Zheng et al., 2019)”.
References:
Chen, S.X.; Xie, Y.Z. On the taxonomic name and species characteristics of Dacus dorlisa. Acta Entomologica Sinica. 1955, 01, 123-126.
Luo, L.Y.; Chen, C.F. Biological characteristics of Dacus dorsalis pupae (Oriental Fruit Fly) on citrus. China Citrus. 1987, (4), 9-10.
Zheng, L.Y., Zhang, Y., Yang, W.Z., Zeng, Y.Y., Jiang, F., Qin, Y.J., Zhang, J.F., Jiang, Z.C., Hu, W.Z., Guo, D.J., et al. New species-specific primers for molecular diagnosis of Bactrocera minax and Bactrocera tsuneonis (Diptera: Tephritidae) in China based on DNA barcodes. Insects, 2019, 10(12), 447. https://doi.org/10.3390/insects10120447
Reviewer 2 Report
Comments and Suggestions for Authors
- This study incorporates a citrus distribution index as one of the environmental variables in the MaxEnt model to predict the potential distribution of two citrus-specific pests. However, B. minax and B. tsuneonis are oligophagous pests highly specialized on citrus; their occurrence is tightly coupled with the presence of citrus hosts, indicating a symbiotic rather than a unidirectional species–environment relationship.
- The authors assert that “future climate warming will lead to a contraction of suitable habitat for both pest species,” yet offer no mechanistic explanation—such as developmental thresholds, diapause duration, or tolerance to climatic extremes. They also ignore the potential for adaptive evolution, natural or human-assisted dispersal (e.g., long-distance transport of infested nursery stock), and instead extrapolate solely from static climate–distribution correlations, rendering the ecological forecasts highly unreliable.
Author Response
Comments 1: This study incorporates a citrus distribution index as one of the environmental variables in the MaxEnt model to predict the potential distribution of two citrus-specific pests. However, B. minax and B. tsuneonis are oligophagous pests highly specialized on citrus; their occurrence is tightly coupled with the presence of citrus hosts, indicating a symbiotic rather than a unidirectional species–environment relationship.
Response 1: Thank you for your professional comments. You are absolutely right in pointing out that including the citrus distribution index in the model for predicting the distribution of its specialist pests may introduce the possibility of a symbiotic relationship rather than a purely unidirectional process of environmental filtering. Our primary motivation for incorporating this index at the initial modeling stage was to better capture, at the landscape scale, the “accessibility” and “realistic likelihood” of pest occurrence. Even if the climate is perfectly suitable, these highly specialized pests are very unlikely to occur in areas where citrus is not cultivated. Our aim was therefore to produce a distribution map that is more aligned with practical management needs, rather than one that only reflects theoretically suitable climatic zones. To assess this potential effect, we have explicitly addressed and discussed it in the discussion and sections of the manuscript in line 395-402 (same as the response 2).
Comments 2: The authors assert that “future climate warming will lead to a contraction of suitable habitat for both pest species,” yet offer no mechanistic explanation—such as developmental thresholds, diapause duration, or tolerance to climatic extremes. They also ignore the potential for adaptive evolution, natural or human-assisted dispersal (e.g., long-distance transport of infested nursery stock), and instead extrapolate solely from static climate–distribution correlations, rendering the ecological forecasts highly unreliable.
Response2: Thank you very much for your insightful comments regarding the lack of mechanistic explanation and the omission of processes such as adaptive evolution and human-mediated dispersal. We fully agree that this is indeed an inherent limitation of correlation-based ecological niche models. In the present study, MaxEnt was employed as an exploratory tool to conduct a preliminary screening based on the correlation between current species distribution and climatic variables. This approach allows identification of potential high-risk areas and key climatic drivers with relatively low data and computational requirements, thereby providing useful guidance for prioritizing regions for subsequent mechanistic research and targeted monitoring efforts. As you are aware, obtaining the detailed physiological parameters you mentioned remains challenging for many species, especially pest insects. Under such circumstances, MaxEnt is widely adopted to generate an initial version of distribution prediction.
We have added a detailed explanation of the limitations of this study at the end of the discussion section in line 395-402 “Nevertheless, this study does not incorporate the potential effects of adaptive evolu-tionary processes, nor does it account for human-mediated dispersal mechanisms, such as long-distance jump dispersal via seedling or plant material transport. The omission of these processes constitutes a significant source of uncertainty in the predictive out-comes and should be considered a priority for refinement in future research, thereby supporting the citrus industry sustainable and resilient development in China“. These should be prioritized in future research.
Reviewer 3 Report
Comments and Suggestions for Authors
The manuscript presents an interesting approach and comprehensively analyzes the potential geographic distribution of Tetradacus subgenus. The results obtained are important in preventing the invasion and spread of B. minax and B. tsuneonis for China's citrus cultivation. The manuscript needs some minor revisions before publishing.
Please check the Scientific names throughout the manuscript.
Scientific names must be written in full as Genus species, Author (Order: Family) in the main title, summary, and where they appear for the first time in body text.
The authors frequently used the subgenus name, Tetradacus, in the manuscript. It is confusing. Please use the names of the species of test insects.
Please also be careful while using abbreviations
For example, in the Abstract line 34
under the curve value (AUC). AUC value
Abbreviations need to be checked and corrected throughout the whole paper.
Also, check the citation rules,
Some references are missing in the manuscript. I highlighted them in a PDF file. Please check
Introduction line 87-91, please rephrase the paragraph and integrate clearly with the earlier sentences.
2.1. Occurrence Record of Tetradacus
Please add a short paragraph about how you separated the two species in the field.
The language is good in general, but some paragraphs need revision and grammatical checking

In general, the language is good, but some paragraphs need revision and grammatical checking in results section
Author Response
Comments 1: The manuscript presents an interesting approach and comprehensively analyzes the potential geographic distribution of Tetradacus subgenus. The results obtained are important in preventing the invasion and spread of B. minax and B. tsuneonis for China's citrus cultivation. The manuscript needs some minor revisions before publishing.
Response 1: We sincerely thank the reviewer for their positive assessment of our work and their valuable feedback. In accordance with the reviewer’s suggestions, we have carefully addressed all the points raised and have made the necessary revisions to improve the manuscript. We believe these changes have further strengthened the paper.
Comments 2: Please check the Scientific names throughout the manuscript. Scientific names must be written in full as Genus species, Author (Order: Family) in the main title, summary, and where they appear for the first time in body text.
Response 2: Thank you for this important reminder regarding the proper formatting of scientific names. We have carefully reviewed the entire manuscript and have now ensured that the scientific names adhere to the specified format. In line 10-11,29-30. “Bactrocera minax (Enderlein) (Diptera: Tephritidae) and Bactrocera tsuneonis (Miyake) (Diptera: Tephritidae)”. In line 338. “Apolygus lucorum (Hemiptera: Miridae)”
Comments 3: The authors frequently used the subgenus name, Tetradacus, in the manuscript. It is confusing. Please use the names of the species of test insects.
Response 3: I apologize for the oversight. We have retained the description of Bactrocera minax and Bactrocera tsuneonis as belonging to the same subgenus of Tetradacus in the abstract (line 27-28,”The Bactrocera minax (Enderlein) (Diptera: Tephritidae) and Bactrocera tsuneonis (Miyake) (Diptera: Tephritidae) are the only members of the subgenus Tetradacus of Bactrocera”) and instruction (line 53-55 “The Tetradacus is the subgenus of Bactrocera (family Tephritidae), widely distributed across tropical and subtropical regions of East Asia . Bactrocera minax and Bactrocera tsuneonis are the only two species classified under Tetradacus.”) and correct all other names to these of the test insects.
Comments 4: Please also be careful while using abbreviations.For example, in the Abstract line 34 under the curve value (AUC). AUC value.Abbreviations need to be checked and corrected throughout the whole paper.
Response 4: I apologize for that. In the abstract section, we have updated“area under the curve (AUC)”,”with an AUC value of 0.969” to “with an area under the curve value (AUC) of 0.969” in line 38, “employed the MaxEnt model” to “employed the Maximum Entropy model (MaxEnt)” in line 34. In addition, we have reviewed all abbreviations in the manuscript to ensure they are defined on first occurrence and remain consistent and standardized in subsequent use.
Comments 5: Also, check the citation rules, some references are missing in the manuscript. I highlighted them in a PDF file. Please check
Response 5: I am sorry for that. We have corrected this.I have inserted all the missing citations that were highlighted in the PDF. And we have verified the formatting of all references against the journal's author guidelines to ensure full compliance with the citation rules.
Comments 6: Introduction line 87-91, please rephrase the paragraph and integrate clearly with the earlier sentences.
Response 6: I sincerely apologize for this. We have improved the paragraph structure to make it more consistent with the context in line 95-100. “For example, Fu et al. predicted that B. minax potential suitable habitats may shift toward the lower–middle Yangtze River Basin under future climates. Similarly, Mao employed the MaxEnt model to predict that, under future climate change scenarios, B. tsuneonis is likely to expand its potential suitable range into northern and western China.”
Comments 7: 2.1. Occurrence Record of Tetradacus. Please add a short paragraph about how you separated the two species in the field.
Response 7: I am sorry for that. In introduction section we add some knowledge and cite relative references in line 68-71. “The two pests share almost identical morphological characteristics and occupy highly overlapping ecological niches. B. tsuneonis can be distinguished from B. minax by its 1-2 pairs of anterior supra-alar seta on the mesonotum (absent in B. minax) and a female ovipositor about half the abdominal length (as long as the abdomen in B. minax) (Chen and Xie, 1955; Luo and Chen, 1987). In addition, the two pests can be accurately identified through DNA barcoding (Zheng et al., 2019)”.
References:
Chen, S.X.; Xie, Y.Z. On the taxonomic name and species characteristics of Dacus dorlisa. Acta Entomologica Sinica. 1955, 01, 123-126.
Luo, L.Y.; Chen, C.F. Biological characteristics of Dacus dorsalis pupae (Oriental Fruit Fly) on citrus. China Citrus. 1987, (4), 9-10.
Zheng, L.Y., Zhang, Y., Yang, W.Z., Zeng, Y.Y., Jiang, F., Qin, Y.J., Zhang, J.F., Jiang, Z.C., Hu, W.Z., Guo, D.J., et al. New species-specific primers for molecular diagnosis of Bactrocera minax and Bactrocera tsuneonis (Diptera: Tephritidae) in China based on DNA barcodes. Insects, 2019, 10(12), 447. https://doi.org/10.3390/insects10120447
Round 2
Reviewer 2 Report
Comments and Suggestions for Authors
Although the authors have addressed some issues from the previous round of review, fundamental flaws remain that preclude acceptance of this manuscript. While the authors claim that incorporating the "citrus distribution index" represents a key innovative contribution, the construction of this variable exhibits critical methodological flaws. First, the citrus distribution data were derived from GBIF occurrence records without validating their spatial accuracy, sampling bias, or completeness. GBIF data for agricultural species often confuse cultivated records with natural distributions; the direct use of kriging interpolation to generate a nationwide continuous raster therefore misclassifies the non-natural distribution of cultivated orchards as native species habitat, violating fundamental assumptions of ecological niche modeling. Second, the environmental variable selection process lacks transparency. The manuscript merely states that variables were removed when Pearson correlation coefficients exceeded 0.8, but does not provide the complete variable correlation matrix or conduct sensitivity tests on how the final variable selection affects model transferability, precluding assessment of overfitting risks. Most critically, among the 134 total occurrence records for both species, 91 are from B. minax and only 43 from B. tsuneonis—the substantial sample size imbalance, coupled with the lack of spatial thinning, likely leads to model overfitting to the distribution pattern of B. minax, while predictions for B. tsuneonis are associated with extremely low confidence. These methodological issues have not been substantively improved in the revised manuscript, undermining the reliability of all subsequent distribution predictions and climate scenario analyses—problems that cannot be rectified through minor revisions.